# Effects of Individualised High Positive End-Expiratory Pressure and Crystalloid Administration on Postoperative Pulmonary Function in Patients Undergoing Robotic-Assisted Radical Prostatectomy: A Prospective Randomised Single-Blinded Pilot Study

**DOI:** 10.3390/jcm12041460

**Published:** 2023-02-12

**Authors:** Sebastian Blecha, Anna Hager, Verena Gross, Timo Seyfried, Florian Zeman, Matthias Lubnow, Maximilian Burger, Michael T. Pawlik

**Affiliations:** 1Department of Anaesthesiology, University Medical Centre Regensburg, Franz-Josef-Strauss-Allee 11, 93053 Regensburg, Germany; 2Department of Anaesthesiology, Caritas St. Josef Medical Centre, University Medical Centre Regensburg, 93053 Regensburg, Germany; 3Centre for Clinical Studies, University Medical Centre Regensburg, 93053 Regensburg, Germany; 4Department of Internal Medicine II, University Medical Centre Regensburg, 93053 Regensburg, Germany; 5Department of Urology, Caritas St. Josef Medical Centre, University Medical Centre Regensburg, 93053 Regensburg, Germany

**Keywords:** fluid management, individual PEEP, pulmonary function, robotic-assisted laparoscopic prostatectomy, steep Trendelenburg position

## Abstract

Objectives: Robotic-assisted laparoscopic prostatectomy (RALP) is typically conducted in steep Trendelenburg position (STP). The aim of the study was to evaluate whether crystalloid administration and individual management of positive end-expiratory pressure (PEEP) improve peri- and post-operative pulmonary function in patients undergoing RALP. Design: Prospective randomised single-centre single-blinded explorative study. Setting: Patients were either allocated to a standard PEEP (5 cmH_2_O) group or an individualised high PEEP group. Furthermore, each group was divided into a liberal and a restrictive crystalloid group (8 vs. 4 mL/kg/h predicted body weight). Individualised PEEP levels were determined by means of preoperative recruitment manoeuvre and PEEP titration in STP. Participants: Informed consent was obtained from 98 patients scheduled for elective RALP. Interventions: The following intraoperative parameters were analysed in each of the four study groups: ventilation setting (peak inspiratory pressure [PIP], plateau pressure, driving pressure [P_driv_], lung compliance [LC] and mechanical power [MP]) and postoperative pulmonary function (bed-side spirometry). The spirometric parameters Tiffeneau index (FEV_1_/FVC ratio) and mean forced expiratory flow (FEF_25–75_) were measured pre- and post-operatively. Data are shown as mean ± standard deviation (SD), and groups were compared with ANOVA. A *p*-value of <0.05 was considered significant. Results: The two individualised high PEEP groups (mean PEEP 15.5 [±1.71 cmH_2_O]) showed intraoperative significantly higher PIP, plateau pressure and MP levels but significantly decreased P_driv_ and increased LC. On the first and second postoperative day, patients with individualised high PEEP levels had a significantly higher mean Tiffeneau index and FEF_25–75_. Perioperative oxygenation and ventilation and postoperative spirometric parameters were not influenced by restrictive or liberal crystalloid infusion in either of the two respective PEEP groups. Conclusions: Individualised high PEEP levels (≥14 cmH_2_O) during RALP improved intraoperative blood oxygenation and resulted in more lung-protective ventilation. Furthermore, postoperative pulmonary function was improved for up to 48 h after surgery in the sum of the two individualised high PEEP groups. Restrictive crystalloid infusion during RALP seemed to have no effect on peri- and post-operative oxygenation and pulmonary function.

## 1. Introduction

Robotic-assisted laparoscopic prostatectomy (RALP) has been the fastest-developing surgical technique for treating prostate carcinoma in recent years. RALP has many advantages over open surgery, including minimal tissue trauma, lower blood loss and transfusion rates, fewer surgical and postoperative complications, earlier postoperative recovery and improved function [1,2,3]. RALP requires pneumoperitoneum and steep Trendelenburg positioning (STP) with a head-down tilt of at least 35 to 45 degrees. The necessity of having to maintain adequate ventilation and normal CO_2_ levels in patients undergoing RALP makes it difficult to avoid high minute ventilation with increased peak inspiratory pressure (PIP) [4,5]. Ventilation decreases lung movement, resulting in elevation of the diaphragm. Many anaesthesiologists increase PEEP to prevent collapse of the airways. The aim of a ’best PEEP concept’ is to detect the optimal individual PEEP level to enable non-harming ventilation for lung compliance (LC) and to minimise the development of mechanical lesions due to ventilation [6].

In the past twenty years, much knowledge has been gained on the lung physiology of both healthy and critically ill patients. Limited high plateau pressure and tidal volume (V_T_), appropriate positive end-expiratory pressure (PEEP), restriction of fluid and adequate driving pressure (P_driv_) in the lung (i.e., plateau pressure—PEEP) have resulted in fewer days of ventilation and better survival rates in many countries, at least in critically ill patients with acute respiratory distress syndrome (ARDS) [7,8,9]. In 2015, Amato et al. described the association of P_driv_ with mortality in patients with ARDS, and this report has led to a different perspective on ventilation during anaesthesia, even in patients without respiratory diseases [10]. Furthermore, protective ventilation with lower V_T_ has been associated with better clinical outcome, even in patients without ARDS [11]. Moreover, stressed lungs are prone to collect fluid in the interstitial space, leading to a decline in oxygen saturation and postoperative lung function tests [12].

Fluid therapy is an important cornerstone in perioperative management, and it may influence clinical outcomes [13]. Fluid overload results in interstitial oedema, increased cardiorespiratory workload, and body weight gain, whereas insufficient fluid loading leads to poor peripheral blood flow and low tissue oxygen delivery. A recent study defined 3 to 4 mL/kg/h balanced crystalloids as receiving restrictive normovolemic therapy in patients undergoing elective open abdominal surgery [14]. In patients undergoing colorectal surgery, a restricted perioperative intravenous fluid regimen reduced postoperative complications [15]. In a recent study of patients undergoing major abdominal surgery, however, a restrictive fluid regimen (6.5 mL/kg/h) was associated with a higher rate of acute kidney injury [16]. The most appropriate fluid regimen for patients undergoing RALP is yet unknown.

In our study, we prospectively analysed patients to assess whether different intraoperative fluid regimens and PEEP settings had an influence on ventilation and oxygenation during RALP and on pulmonary function up to 48 h after surgery. We hypothesised that the combination of individualised high PEEP values and restrictive administration of crystalloids increases postoperative pulmonary function and improves perioperative oxygenation and ventilation in patients undergoing RALP.

## 2. Methods

### 2.1. Study Design

This prospective randomised single-centre single-blinded study was approved by the local institutional review board (Protocol no. 18-1224-101, approved on 12 December 2018) and registered with the German Clinical Trials Register (DRKS00016887, prospectively registered on 7 March 2019). Informed consent was obtained from study patients scheduled for elective prostatectomy at the Department of Urology at the Caritas St. Josef University Medical Centre Regensburg. All patients were recruited between March 2019 and October 2019. Main exclusion criteria were pre-existing pulmonary disease (bronchial asthma, chronic obstructive pulmonary disease [COPD]), age > 80 years, body mass index [BMI] > 35 kg/m², American Society of Anaesthesia physical status > III, known cardiac and renal insufficiency and pulmonary hypertension). Randomisation was conducted by the local Centre for Clinical Studies. A randomisation list was generated using SAS 9.4 (SAS Institute Inc., Cary, NC, USA) and the procedure plan (block-wise randomisation with a block length of 8). An envelope was opened by the principal investigator (MP) after the patient had signed the informed consent.

### 2.2. Patient Cohort

The patients were preoperatively randomised into one of the four study groups (Figure 1): standard (5 cmH_2_O) or individualised PEEP with administration of either restrictive (4 mL/kg/h predicted body weight [PBW] up to one hour after RALP) or liberal (8 mL/kg/h PBW up to one hour after RALP) crystalloids. The study preliminarily included 104 patients. Six patients were excluded from analysis because of missing data or study protocol violations (missing of ventilation and/or spirometry settings [n = 5] or retroactive withdrawal of consent [n = 1]).

### 2.3. Anaesthesia Protocol and Surgical Technique

A standardised anaesthesia protocol for drug administration during RALP was exclusively conducted by three anaesthesiologists throughout the entire study. Drug dosing was based on the calculated PBW (PBW formula: PBW [men] = 50 + 0.91 × [cm of height—152.4] in kg) [17]. Anaesthesia was induced with sufentanil (initial 0.5 µg/kg bolus), propofol (2–3 mg/kg) and atracurium (0.5 mg/kg). After tracheal intubation with a 7.5 mm or an 8.0 mm endotracheal tube, anaesthesia was maintained with sufentanil (repetition of 10 µg every 30 to 45 min until 30 min before the end of surgery) and propofol (4–6 mg/kg/h) as total intravenous anaesthesia (TIVA). TIVA was used as standard anaesthesia for RALP to minimise the influence on pulmonary function by volatile anaesthetics. A Bispectral Index™ (BIS Vista Monitor, Aspect Medical, Germany) between 40 and 50 was upheld during anaesthesia. Invasive blood pressure was measured directly after the induction of anaesthesia using a radial artery catheter. All patients were placed by default in STP to check the correct positioning and solid fixation on the operating table. In this context, the individualised PEEP group received one recruitment manoeuvre (RM) followed by a decremental PEEP titration in STP.

The RM was performed in volume-controlled mode and consisted of 10 respiratory cycles with a PEEP level of 22 cmH_2_O, a peak inspiratory pressure of 40 cmH_2_O and a ventilation frequency of 6 breaths per min with an I/E of 1:2. For the decremental PEEP titration PEEP was set to 20 cmH_2_O and decreased stepwise by 2 cmH_2_O every 3 min. At each PEEP step, the best lung compliance value was observed, and this individual PEEP level was maintained throughout mechanical ventilation during surgery. No RM were employed in the standard PEEP group.

During RALP, the target values for SpO_2_ were defined as higher than 92% and those of mean arterial pressure (MAP) as 60 mmHg. Otherwise, FiO_2_ or noradrenaline concentration was adapted. All patients received volume-controlled ventilation with PEEP according to the levels predefined for the respective group, using an inspiration-to-expiration ratio of 1:1, a basic respiratory rate of 12 and a constant V_T_ of 7–8 mL/kg PBW. At the beginning of RALP, pneumoperitoneum was created by intraperitoneal insufflation of CO_2_ to a standard value of 15 mmHg with the patient in supine position. Subsequently, each patient was consequently placed in 45 degrees STP. Surgery was exclusively conducted by three urologists. Neuromuscular transmission was monitored with a peripheral nerve stimulator to maintain one twitch of the train-of-four (TOF). Relaxation with rocuronium was finished 45 min before the end of surgery. For extubation, the TOF ratio had to be >0.9 at the end of RALP. Individualised high PEEP values were reduced to 8 cmH_2_O at the end of surgery after positioning the patient in supine position prior to extubation.

### 2.4. Measurement of Perioperative Ventilation and Blood Oxygenation

All patients were ventilated with the Perseus^®^ Ventilator (Dräger Medical, Lübeck, Germany). Lung parameters and blood oxygenation were measured in each patient at predefined time points (Figure 1). The following respiratory parameters were documented during mechanical ventilation: fraction of inspired oxygen (F_i_O_2_), minute volume (MV), respiratory rate, tidal volume (V_T_), peak inspiratory pressure (PIP), plateau pressure, PEEP, dynamic LC and end-tidal CO_2_. Blood gases were analysed with the radiometer ABL 800 Flex (Radiometer Medical, Copenhagen, Denmark). The following parameters were investigated: pH value, partial pressure of arterial oxygen (P_a_O_2_), partial pressure of arterial carbon dioxide (PaCO_2_), P_a_O_2_/F_i_O_2_ ratio (P/F ratio), peripheral oxygen saturation (SpO_2_), arterial oxygen (SaO_2_) and base excess (BE). Additionally, we calculated the driving pressure (P_driv_) as the difference between plateau pressure and PEEP and the mechanical power (MP) with the following formula: MP (in J/min) =0.098 × V_T_ × RR × (PIP − 0.5 × P_driv_) [18,19].

### 2.5. Measurement of Postoperative Pulmonary Function, Body Weight and Brain Natriuretic Peptide

Spirometry was carried out with the Vitalograph Micro^®^ (Vitalograph GmbH, Hamburg, Germany) with the patient in sitting position at the following four time points: on the day before surgery, 60 min after extubation in the recovery room and on the first and second postoperative day. Each spirometric measurement was observed by two physicians (VG, AH), who had received technical instruction by the manufacturer and had been briefed in using the spirometer by a pulmonologist (ML). The physicians were blinded for the pulmonary function measurement, i.e., they did not know into which group the patients had been randomised. The following parameters were measured with the spirometer: vital capacity (VC), forced vital capacity (FVC), forced expiratory volume in one second (FEV_1_), the Tiffeneau index (FEV_1_/FVC ratio), peak expiratory flow (PEF), forced expiratory flow after one quarter of FVC (FEF_25_), forced expiratory flow after half of FVC (FEF_50_), forced expiratory flow after three quarter of FVC (FEF_75_) and mean forced expiratory flow (FEF_25–75_). Additionally, body weight and brain natriuretic peptide (BNP) were established preoperatively and on the first and second postoperative day. A normal BNP level was defined as less than 35 pg/mL [20].

### 2.6. Study Aims

The primary aim of the study was to investigate the dynamics of postoperative spirometric parameters (VC, FVC, FEV_1_, FEV_1_/FVC ratio, PEF, FEF_25_, FEF_50_, FEF_75_ and FEF_25-75_), differentiated between the four different study groups, from the recovery room up to the second postoperative day. In an additional sub-analysis, spirometric parameters of the sum of the two standard PEEP groups were compared with those of the sum of the individualised PEEP groups, or the sum of the two restrictive volume groups with those of the sum of the liberal volume groups. Secondary aims included parameters regarding perioperative ventilation (F_i_O_2_, MV, V_T,_ respiratory rate, PIP, plateau pressure, PEEP, P_driv_, dynamic LC, MP and end-tidal CO_2_) and blood oxygenation (P_a_O_2_, PaCO_2_, SaO_2_, BE, P/F ratio) as well as changes in body weight and BNP.

### 2.7. Statistics

#### 2.7.1. Sample Size Considerations

This study was designed as a pilot study with several primary variables to get insights about postoperative pulmonary function depending on standard or individualised PEEP and on the administration of restrictive or liberal crystalloids. Thus, no a priori sample size calculation could be performed. Because we expected small to medium effect sizes, a sample size of n = 25 per group (n_total_ = 100) was targeted to obtain rather robust effect estimates in all four groups for further studies [21,22].

#### 2.7.2. Statistical Methods

Data are shown as mean ± SD for continuous variables and as absolute and relative frequencies for categorical variables. All continuous variables were compared between two or more groups by using an analysis of variance (ANOVA). All reported *p*-values are two-sided, and a *p*-value of 0.05 was considered the threshold of statistical significance. Due to the explorative nature of this study, no adjustment for multiple testing was done. All analyses were carried out using the software R (Version 4.0.4, www.r-project.org (accessed on 11 April 2022)) by a biomathematician and statistician (FZ).

### 2.8. Patient and Public Involvement

Patients or the public were not involved in the design, or conduct, or reporting, or dissemination plans of our research.

## 3. Results

Out of 104 patients, who had given their informed consent, 98 male patients were eligible for statistical analysis. Patient characteristics, duration of surgery and anaesthesia are listed in Table 1. Mean age was 64 years (±7 years), mean BMI 26.9 kg/m² (±3.3 kg/m²) and mean PBW 72.6 kg (±4.7 kg). Mean duration of surgery was 160 min (±36 min) and mean duration of anaesthesia 228 min (±40 min). The four study groups did not differ in any of the described parameters, and the infused crystalloid volume in each group complied with the study protocol. None of the patients in this study needed postoperative intensive care treatment. Length of stay was 7 days (range 6 to 8) in the standard PEEP groups and 7 days (range 6 to 7) in the individualised PEEP groups without any statistical difference between the groups (*p* = 0.25).

### 3.1. Perioperative Ventilation and Haemodynamic and Blood Oxygenation

The respiratory parameters of the four groups are depicted in Figure 2 and Appendix A. The two individualised PEEP groups had a combined mean PEEP level of 15.5 cmH_2_O (±1.71 cmH_2_O) and significantly higher plateau pressure and MP levels but significantly lower P_driv_ levels than the two standard PEEP groups (5 cmH_2_O) at all measuring time points during RALP. Perioperative P_driv_ levels were lowest in the individualised PEEP group with restrictive crystalloid infusion. Dynamic LC and blood oxygenation measured by the P/F ratio were significantly higher in the two individualised PEEP groups. The gap between PaCO_2_ and end-tidal CO_2_ ranged between 3 to 5 mmHg without any significance between the four study groups and the different time points of RALP.

The two groups receiving different crystalloid infusion did not differ in plateau pressure, MP, PEEP and P_driv_ levels, dynamic LC and P/F ratio during RALP; only at T4 were the MAP values significantly lower in the group receiving restrictive crystalloid infusion than in the group receiving liberal crystalloid infusion (88.5 vs. 95.1 mmHg, *p* = 0.028).

### 3.2. Postoperative Pulmonary Function, Body Weight and Brain Natriuretic Peptide

The results of pre- and post-operative spirometry are shown in Table 2. All spirometric parameters measured in the recovery room were decreased in comparison to preoperative values, but there were no significant differences between the four study groups. For example, FEV1 in the recovery room was decreased by 22% (0.69 litre) and FVC by 18.6% (0.79 litre) compared to preoperative values across all four study groups. Up to 48 h after RALP, the spirometric parameters (Tiffeneau index, FEF_25_, FEF_50_, FEF_75_ and FEF_25-75_) increased again but not reached completely the preoperative values. Patients with individualised PEEP tended to have higher levels, but there were no significant differences between the four groups.

A comparison of the sum of the two standard PEEP groups with the sum of the individualised PEEP groups (Appendix A) showed that patients with individualised PEEP had a significantly higher mean Tiffeneau index on the first and second postoperative day than patients with standard PEEP (1st day: 77.5% vs. 73.6%, *p* = 0.014; 2nd day: 76.5% vs. 72.7%, *p* = 0.021) as well as higher FEF_25–75_ (1st day: 2.41 vs. 1.95 litre/s, *p* = 0.009; 2nd day: 2.45 vs. 2.07 litre/s, *p* = 0.033). Postoperative spirometric parameters were not influenced by restrictive or liberal crystalloid infusion, neither in the four study groups (Table 2) nor in the two added-up crystalloid groups (Appendix A).

The preoperative values and postoperative changes in body weight and BNP levels of each study group are shown in Appendix A. In all study patients, mean body weight was increased on the first postoperative day (+1.5 kg or +1.8%) and decreased on the second postoperative day (+0.4 kg or +0.4%) compared to preoperative values. Mean BNP levels were elevated in all study patients before surgery and on the first postoperative day (+30.0 pg/mL or +64.1%). On the second postoperative day, mean BNP levels had decreased below preoperative values in each study group (−5.9 pg/mL or −12.2%). The study groups did not show any significant differences.

## 4. Discussion

The main findings were: (1) the application of individualised higher PEEP significantly increased plateau pressure, MP, perioperative oxygenation (P/F ratio) and LC and significantly decreased P_driv_, and (2) spirometric parameters (especially the Tiffeneau index and FEF_25–75_) were significantly increased in the sum of patients with perioperatively higher PEEP levels up to the second postoperative day. Secondary findings were: (1) neither restrictive nor liberal crystalloid infusion during surgery had a significant influence on postoperative pulmonary function, and (2) the study groups did not significantly differ in pre-and post-operative BNP levels and body weight.

### 4.1. Perioperative Ventilation and Blood Oxygenation

In this study, individualised higher PEEP levels resulted in significantly better blood oxygenation (P/F ratio) and lung-protective ventilation (increased LC and reduced P_driv_) but also in significantly higher calculated MP on the lungs, probably due to the application of significantly higher PIP in the individualised high PEEP groups. The promising idea to combine several variables related to ventilator-induced lung injury (VILI) in the equation of MP [23] has still several limitations, especially the problem of applying appropriate PEEP, and has some qualitative disagreements with the clinical data available on VILI [24], as seen in our study. On the one hand, high MP applied to lungs may be harmful, as seen in our individualised PEEP group. On the other hand, we observed a decrease in P_driv_ and better oxygenation, LC and postoperative spirometric lung function, although these findings should not be overestimated. Nestler et al. found higher PEEP levels, improved blood oxygenation and significantly reduced P_driv_ in obese patients undergoing elective laparoscopic bariatric surgery in general anaesthesia, but these differences had not persisted after extubation [25]. Inadequately low PEEP values during laparoscopic surgery (in STP) facilitate the development of atelectasis and decrease pulmonary ventilation and oxygenation. Two recent studies using electrical impedance tomography during RALP showed that PEEP with 14 to 15 cmH_2_O improves oxygenation in non-obese patients [26,27].

VT changes during robot-assisted laparoscopic surgery result in fluid shifts, which are detected by pulse pressure variation and stroke volume variation [28]. Otherwise, passive leg raising does not predict fluid responsiveness in patients with intra-abdominal pressure > 16 mmHg [29]. Neither restrictive nor liberal crystalloid management influenced perioperative ventilation and oxygenation in our study. One possible explanation for this result is that abdominal laparoscopic surgery may potentially influence dynamic variables through direct mechanical effects and the autonomic nervous system, thus confounding the effects of different fluid challenges. Furthermore, pneumoperitoneum probably affects the volume mobilised by head-down tilts. Only MAP in longer duration of pneumoperitoneum and STP (T4) were significantly reduced in both restrictive groups compared to the liberal crystalloid groups. This factor should be taken into consideration in longer duration of RALP. The higher demand for noradrenaline in the higher PEEP groups at the end of surgery (T5) may be explained by the termination of pneumoperitoneum and STP, which results in a fall in blood pressure due to lower-positioned legs under high PEEP.

### 4.2. Postoperative Pulmonary Function, Body Weight and Brain Natriuretic Peptide

The combination of STP and pneumoperitoneum for 2 to 3 h may cause oedema of the upper airway and reduce pulmonary compliance [4,30]. In all groups of our study, spirometric parameters after RALP were decreased compared to preoperative measurements and showed the lowest values in the recovery room. One possible explanation may be reduced alertness and postoperative pain, which impedes breathing up to 1 h after surgery. Different intraoperative ventilator and fluid management had no influence at that time point. Kilic and colleagues investigated the effects of STP on the intra- and extrathoracic airways in patients with and without COPD who undergo RALP. In patients without COPD, VC and FEV_1_ were reduced after RALP, but levels had recovered within 5 days; in contrast, in patients with COPD, this reduction lasted more than 5 days [31]. Our study patients in the group with higher PEEP and preoperatively normal lung function showed a significantly better Tiffeneau index and higher FEF_25-75_ values on the first and second postoperative day. Because a decreased Tiffeneau index is a sign of increased bronchoconstriction in the larger airways, decreased FEF_25-75_ values represent higher flow resistance of the small airways of the lung. This may be the result of increased capillary leakage during STP, leading to increased flow resistance; individualised PEEP and restrictive crystalloid management may prevent the influx of fluid into the interstitial space in the lungs.

In our study, the volume of administered crystalloids did not influence postoperative pulmonary function. So far, no previous study has investigated the impact of different fluid regimen on intra- and post-operative pulmonary function during RALP. Piegeler et al. evaluated the influence of intraoperative fluid administration on postoperative complications (urologic complications and length of hospitalisation) in 182 patients undergoing RALP. High amounts of crystalloids were associated with an increased incidence of anastomotic leakage, especially in older patients, but had no effect on the length of hospitalisation [32].

Measurement of circulating levels of BNP is a diagnostic tool for identifying patients with elevated ventricular filling pressure who are likely to develop symptoms of heart failure [33,34]. STP and pneumoperitoneum during RALP caused a 2- to 3-fold increase in cardiac preload and may have resulted in acute heart failure [30,35,36]. Several previous studies have described perioperative cardiac ischemia during and after RALP [37,38]. In our study, no postoperative cardiac complications were observed. Mean BNP levels and body weight increased on the first postoperative day but had returned to normal on the second postoperative day, independent of PEEP and volume administration. To the best of our knowledge, no previous study has yet investigated changes in BNP and bodyweight in patients undergoing RALP.

### 4.3. Strengths and Limitations

The strengths of the present study are its prospective randomised single-blinded design and the measurement of pulmonary function up to 48 h after RALP in a large number of patients. Moreover, the study was performed only by a few doctors within a very short time, which allowed a very strict standardisation of the study process and accurate measurements. Taking into account the demanding requirements of mechanical ventilation during STP, we have added MP calculations during laparoscopic surgery to the existing body of knowledge, for the first time. Nevertheless, spirometry is a measurement method that strongly depends on patient cooperation. Considering the exclusion criteria, the generalization of our results to those patients could be limited. Because of the explorative nature of this study and the small sample size in the subgroups, no statement can be made on the effects of liberal or restrictive crystalloid regimens.

## 5. Conclusions

Patients with individualised higher PEEP levels (>14 cmH_2_O) compared to standard PEEP (5 cmH_2_O) during RALP showed improved blood oxygenation (P/F ratio) and LC as well as decreased P_driv_, which improved postoperative pulmonary function up to 48 h after surgery in the sum of the individualised high PEEP groups. A preoperative individual PEEP titration to define the best ventilation setting could result into minimize lung injuries during RALP and should be researched in future studies. In our study, restrictive crystalloid infusion during RALP seems to have no effect on peri- and post-operative oxygenation and pulmonary function.

## Figures and Tables

**Figure 1 jcm-12-01460-f001:**
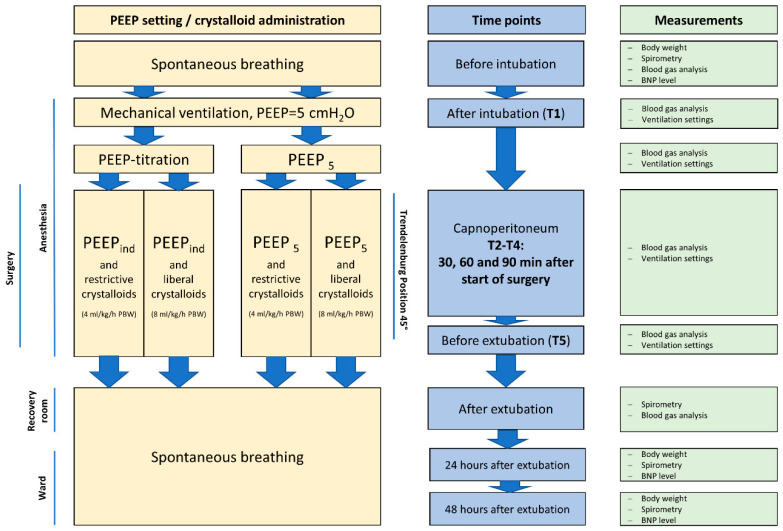
Schematic diagram of the protocol and interventions for the study groups.

**Figure 2 jcm-12-01460-f002:**
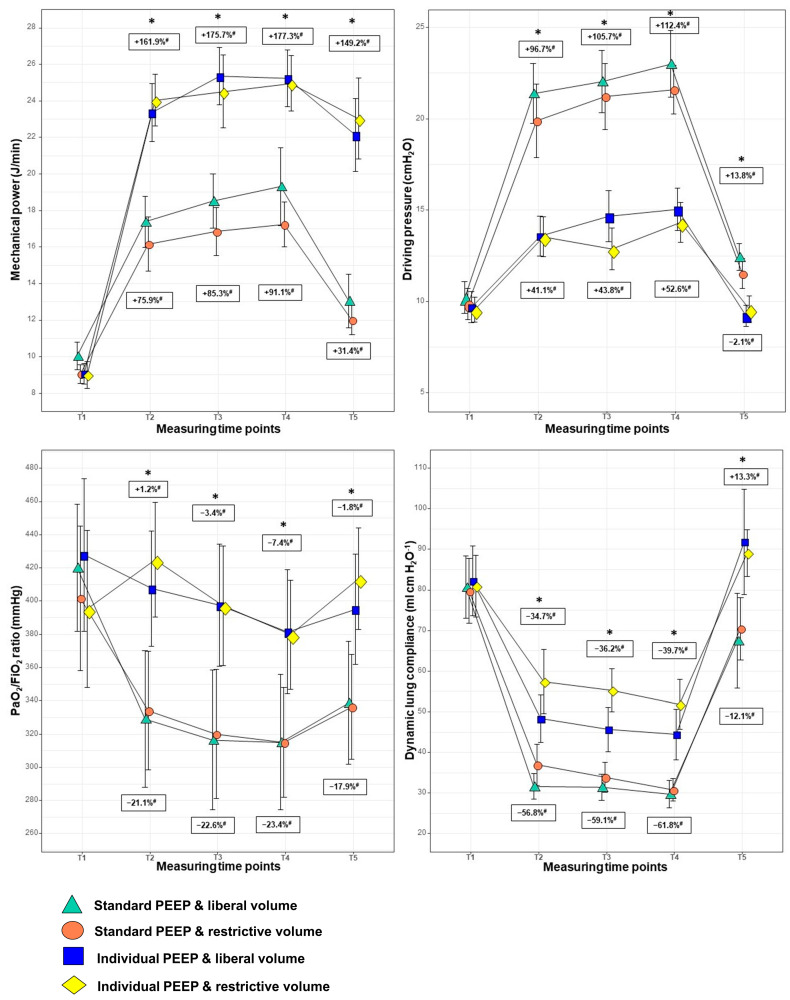
Differences in mechanical ventilation parameters between the study groups during robotic-assisted laparoscopic prostatectomy (n = 98) Notes: * *p* < 0.05; ^#^ absolute changes (%) in the added-up standard or individualised PEEP groups compared to T1; entries depict the mean (SD), and *p*-values compare the four treatment arms at each timepoint using an ANOVA; FiO_2_: fraction of inspired oxygen; PaO_2_: partial pressure of oxygen in arterial blood; T1: 5 min after intubation in supine position, T2: 30 min after the start of pneumoperitoneum and Trendelenburg position, T3: 60 min after the start of pneumoperitoneum and Trendelenburg position, T4: 90 min after the start of pneumoperitoneum and Trendelenburg position, T5: before extubation in supine position; mechanical power: calculated with the following formula: mechanical power = 0.098 × tidal volume × respiratory rate × (PIP − 0.5 × driving pressure).

**Table 1 jcm-12-01460-t001:** Baseline patient and surgical characteristics (n = 98).

	PEEP_5_ and Liberal Volume (n = 23)	PEEP_5_ and Restrictive Volume (n = 27)	PEEP_IND_ and Liberal Volume (n = 24)	PEEP_IND_ and Restrictive Volume (n = 24)	*p*-Value
**Age** (years)	65 (±6)	63 (±7)	64 (±7)	64 (±9)	0.846
**Height** (cm)	177 (±6)	178 (±4)	175 (±5)	177 (±5)	0.193
**Weight** (kg)	89 (±15)	85 (±10)	82 (±10)	82 (±11)	0.143
**BMI** (kg/m²)	28.2 (±3.5)	26.6 (±3.2)	26.8 (±3.4)	26.13 (±3.0)	0.177
**PBW** (kg)	72.8 (±5.2)	73.7 (±3.9)	70.9 (±4.8)	72.7 (±4.5)	0.193
**Volume/PBW *** (mL/kg)	30.8 (±2.3)	15.5 (±2.6)	29.6 (±2.8)	15.8 (±2.8)	**<0.001**
**Volume/PBW/End of recovery room observation *** (mL/kg/h)	8.1 (±1.2)	4.2 (±1.0)	7.8 (±1.3)	4.2 (±1.2)	**<0.001**
**Duration of surgery** (min)	165 (± 35)	155 (± 33)	165 (± 43)	153 (± 35)	0.532
**Duration of anaesthesia** (min)	228 (± 41)	221 (± 39)	235 (± 43)	228 (± 39)	0.703

Notes: Entries depict the mean (±SD), and *p*-values compare the four arms using repeated-measures ANOVA; BMI: body mass index; PBW: predicted body weight; PEEP_IND_: designation for the intervention arm, in which individual PEEP was determined after recruitment manoeuvre and PEEP titration; PEEP_5_: designation for the control arm, in which standard PEEP of 5 cmH_2_O was used; restrictive volume: 4 mL/kg/ PBW crystalloid solution up to one hour after RALP; liberal volume: 8 mL/kg/ PBW crystalloid solution up to one hour after RALP, * up to one hour after RALP.

**Table 2 jcm-12-01460-t002:** Comparison of spirometric parameters between four study groups at four time points: the day before surgery, in the recovery room and on the first and second postoperative day (n = 98).

Parameters	Time Point	PEEP_5_ and Liberal Volume (n = 23)		PEEP_5_ and Restrictive Volume (n = 27)		PEEP_IND_ and Liberal Volume (n = 24)		PEEP_IND_ and Restrictive Volume (n = 24)		*p*-Value
**FVC** (L)	Preoperative day	4.40 (±0.83)		4.25 (±0.84)		4.02 (±0.61)		4.30 (±0.63)		0.339
	Recovery room	3.45 (±0.91)	−21.6% ^#^	3.46 (±0.89)	−18.6% ^#^	3.48 (±0.78)	−13.4% ^#^	3.42 (±0.74)	−20.5% ^#^	0.995
	Postoperative day 1	3.39 (±0.86)	−23.0% ^#^	3.53 (±0.78)	−16.9% ^#^	3.48 (±0.71)	−13.4% ^#^	3.53 (±0.73)	−17.9% ^#^	0.912
	Postoperative day 2	3.82 (±0.80)	−13.2% ^#^	3.77 (±0.81)	−11.3% ^#^	3.61 (±0.67)	−10.2% ^#^	3.79 (±0.66)	−11.9% ^#^	0.761
**FEV1** (L)	Preoperative day	3.18 (±0.65)		3.13 (±0.58)		3.05 (±0.6)		3.21 (±0.44)		0.786
	Recovery room	2.43 (±0.61)	−23.6% ^#^	2.42 (±0.67)	−22.7% ^#^	2.53 (±0.62)	−17.1% ^#^	2.44 (±0.59)	−24.0% ^#^	0.913
	Postoperative day 1	2.51 (±0.66)	−21.1% ^#^	2.56 (±0.59)	−18.2% ^#^	2.7 (±0.57)	−11.5% ^#^	2.72 (±0.55)	−15.3% ^#^	0.548
	Postoperative day 2	2.8 (±0.6)	−11.9% ^#^	2.69 (±0.61)	−14.1% ^#^	2.77 (±0.56)	−9.2% ^#^	2.89 (±0.54)	−10.0% ^#^	0.677
**FEV_1_/FVC ratio** **(Tiffenau index)** (%)	Preoperative day	72.46 (±8.13)		74.16 (±7.08)		75.82 (±9.83)		74.8 (±6.04)		0.522
	Recovery room	71.44 (±9.71)	−1.4% ^#^	70.05 (±10.68)	−5,5% ^#^	72.81 (±9.42)	−3.9% ^#^	71.24 (±7.57)	−4.8% ^#^	0.781
	Postoperative day 1	74.88 (±8.99)	+3.3% ^#^	72.52 (±8.65)	−2.2% ^#^	77.73 (±5.96)	+2.5% ^#^	77.37 (±7.35)	+3.4% ^#^	0.069
	Postoperative day 2	73.67 (±7.55)	+1.7% ^#^	71.99 (±10.53)	−2.9% ^#^	76.76 (±5.98)	+1.2% ^#^	76.22 (±6.41)	+1.9% ^#^	0.119
**FEF_25_** (L/s)	Preoperative day	6.31 (±1.85)		6.39 (±1.69)		6.35 (±1.92)		6.5 (±1.95)		0.987
	Recovery room	4.66 (±1.65)	−26.1% ^#^	4.38 (±1.89)	−31.5% ^#^	4.62 (±1.82)	−27.2% ^#^	4.06 (±1.35)	−37.5% ^#^	0.589
	Postoperative day 1	4.96 (±1.59)	−21.4% ^#^	5.02 (±2.14)	−21.4% ^#^	5.94 (±1.83)	−6.5% ^#^	5.42 (±1.4)	−16.6% ^#^	0.201
	Postoperative day 2	5.25 (±1.66)	−16.8% ^#^	5.29 (±2.03)	−17.2% ^#^	5.70 (±1.78)	−10.2% ^#^	5.74 (±1.67)	−11.7% ^#^	0.683
**FEF_50_** (L/s)	Preoperative day	3.39 (±1.55)		3.39 (±1.15)		3.65 (±1.23)		3.89 (±1.55)		0.525
	Recovery room	2.58 (±1.19)	−23.9% ^#^	2.56 (±1.21)	−24.5% ^#^	2.85 (±1.09)	−21.9% ^#^	2.48 (±1.03)	−36.2% ^#^	0.698
	Postoperative day 1	**2.80 ** **(±1.17)**	**−17.4% ^#^**	**2.67 ** **(±0.97)**	**−21.2% ^#^**	**3.41 ** **(±1.05)**	**−6.6% ^#^**	**3.32 ** **(±1.20)**	**−14.7% ^#^**	**0.043**
	Postoperative day 2	2.92 (±1.21)	−13.9% ^#^	2.86 (±1.05)	−15.6% ^#^	3.47 (±1.23)	−4.9% ^#^	3.51 (±1.49)	−9.8% ^#^	0.136
**FEF_75_** (L/s)	Preoperative day	0.89 (±0.58)		0.89 (±0.41)		1.10 (±0.46)		1.0 (±0.45)		0.346
	Recovery room	0.72 (±0.37)	−19.1% ^#^	0.72 (±0.37)	−19.1% ^#^	0.80 (±0.34)	−27.3% ^#^	0.89 (±0.48)	−11.0% ^#^	0.382
	Postoperative day 1	0.78 (±0.47)	−12.4% ^#^	0.79 (±0.41)	−11.2% ^#^	0.93 (±0.39)	−15.5% ^#^	0.99 (±0.62)	−1.0% ^#^	0.302
	Postoperative day 2	0.83 (±0.45)	−6.7% ^#^	0.79 (±0.46)	−11.2% ^#^	0.91 (±0.29)	−17.3% ^#^	0.92 (±0.52)	−8.0% ^#^	0.664

Notes: *p* < 0.05: in bold; entries depict the mean (±SD), and *p*-values compare the four arms using repeated-measures ANOVA; ^#^ absolute changes (%) compared to preoperative values; FEF_25:_ forced expiratory flow after one quarter of FVC; FEF_50:_ forced expiratory flow after half of FVC; FEF_75:_ forced expiratory flow after three quarters of FVC; FEV_1:_ forced expiratory volume in one second; FVC_:_ forced vital capacity; PEEP: positive end-expiratory pressure.

## Data Availability

The datasets supporting the conclusions of this article are included within the article and its Appendix A.

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
