# Peer review of "Effects of Individualised High Positive End-Expiratory Pressure and Crystalloid Administration on Postoperative Pulmonary Function in Patients Undergoing Robotic-Assisted Radical Prostatectomy: A Prospective Randomised Single-Blinded Pilot Study"

_jcm, 2023, doi:10.3390/jcm12041460_

Round 1

Reviewer 1 Report

please see word doc. attached

Reviewer 2 Report

Thanks for giving me the opportunity to review this manuscript.

The search for a tool to individualize PEEP based on patients' individual response is important.  In acute respiratory distress syndrome (ARDS), response to positive end-expiratory pressure (PEEP) is variable according to different degrees of lung recruitability. In my opinion, in healthy patients, like in this study, the recruitability is normal and not affected by any disease, so that during  the pneumoperitoneum induction the abdominal pressure abruptly increase reducing lung compliance.

I read with interest this manuscript, but immediately it raised major concerns regarding the background not updated, the methodology used, the description of results and tables so full and less understandable.

I try to list some of them:

1.       The trial is not registered and sample size was not calculated in order to know if the primary outcome they want to investigate is powerful or not.

2.       Recruitment manoeuvres were not reported clearly. However, they based the calculation of the best PEEP using lung compliance correctly.

3.       In the ventilation section the author focused on peak inspiratory pressure (PIP) that we well know is not important because the true pressure in the alveoli is the plateau pressure. They correctly reported the importance of driving pressure in their results.

4.       Findings should better written down, for example that the high PEEP groups had significantly lower driving pressure meaning that they had recruited some part of lung so that the plateaux pressure and driving pressure were reduced, dynamic compliance and P/F ratio were increased.  PIP and MP are marginal.

5.       Tables are too full of data and not focused on reporting the most important ones. And, reading the main text results, the readers run the risk of getting lost.

Round 2

Reviewer 2 Report

Even if not fully, the authors answered to my comments and made changes in the manuscript.